# Bone Remodeling in Children with Acute Lymphoblastic Leukemia: A Two-Year Prospective Longitudinal Study

**DOI:** 10.3390/ijms26094307

**Published:** 2025-05-01

**Authors:** Paola Muggeo, Massimo Grassi, Vito D’Ascanio, Jessica Forte, Vincenzo Brescia, Francesca Di Serio, Laura Piacente, Paola Giordano, Nicola Santoro, Maria Felicia Faienza

**Affiliations:** 1Department of Pediatric Oncology and Hematology, University Hospital of Policlinico, 70124 Bari, Italy; grassimassimo@hotmail.it (M.G.); nico.santoro1956@libero.it (N.S.); 2Institute of Sciences of Food Production (ISPA), Italian National Research Council (CNR), 70126 Bari, Italy; vito.dascanio@cnr.it; 3Pediatric Department, Ospedale Della Murgia “F. Perinei”, 70022 Altamura, Italy; jessicaforte@hotmail.it; 4Clinical Pathology Unit, AOU Policlinico Consorziale di Bari-Ospedale Giovanni XXIII, 70124 Bari, Italy; bresciavincenzo58@gmail.com (V.B.); francesca.diserio@policlinico.ba.it (F.D.S.); 5Pediatric Unit, Department of Precision and Regenerative Medicine and Ionian Area (DiMePre-J), Medical School, University of Bari “Aldo Moro”, Piazza G. Cesare 11, 70124 Bari, Italy; laura.piacente@uniba.it (L.P.); mariafelicia.faienza@uniba.it (M.F.F.); 6Interdisciplinary Department of Medicine, University of Bari “Aldo Moro”, 70124 Bari, Italy; paola.giordano@uniba.it

**Keywords:** acute lymhpoblastic leukemia (ALL), children, bone remodeling, bone turnover, C-terminal-telopeptide-type-I-collagen (CTX), osteocalcin (OC), procollagen-type-I-N-terminal-propeptide (P1NP)

## Abstract

Childhood leukemia survivors are at risk of long-term complications. Data on bone remodeling in childhood acute lymphoblastic leukemia (ALL) are limited. This 2-year prospective longitudinal study investigated bone remodeling and bone turnover markers at diagnosis, during treatment, and until stopping treatment, in ALL patients < 18 years, to clarify the influence of leukemia itself and/or chemotherapy on bone. Methods: A total of 22 ALL children (12 males, age 5.5 ± 3.6 years) underwent blood sampling at the 5 time point (T0−T4). Osteoprotegerin (OPG), receptor-activator-NF-B-ligand (RANKL), osteocalcin (OC), C-terminal-telopeptide-type-I-collagen (CTX), bone-alkaline-phosphatase (bALP), tartrate-resistant acid-phosphatase-5b (TRACP5b), procollagen-type-I-N-terminal-propeptide (P1NP), Dickkopf-1 (DKK-1), and sclerostin were assessed. Data from patients at T0 were compared to a control group of healthy children. We used the principal component analysis (PCA) for statistics. Results: Levels of CTX, OC, P1NP, and bALP resulted lower in ALL children than controls (*p* = 0.009 for CTX and *p* < 0.001 for the others), also DKK1 and sclerostin (*p* < 0.0001 and *p* = 0.023). RANKL ed OPG were higher in patients. During T0−T4, CTX, OC, P1NP, TRACP5b, and bALP showed a significant increase, in particular at T0−T1 (end-of-induction). Less evident changes were detected onwards. Conclusions: The onset of leukemia has been revealed as a key point in determining a slowing of bone remodeling in ALL children.

## 1. Introduction

Improved results in the treatment of childhood acute lymphoblastic leukemia (ALL), thanks to risk-adapted multiagent antileukemic therapy and supportive care, have led to about 90% of a 5-year survival rate [1,2]. However, the growing population of childhood leukemia survivors is at risk of long-term complications, due to the disease itself and to the treatment [3]. Drug exposure is the major determinant of late effects in childhood cancer survivors, including neurologic conditions [4], cardiometabolic alterations [5,6,7], and impact on skeletal mineralization [8,9,10].

Glucocorticoids (GCs) represent a key drug for the treatment of ALL, as both prednisone and dexamethasone, used during the induction and reinduction phases of treatment and in the consolidation phase of high-risk patients, have a crucial role. GCs promote apoptosis of osteoblasts and osteocytes and increase bone resorption, decreasing bone formation [11,12]. Furthermore, GCs influence the serum levels of calcium, reducing intestinal absorption and increasing renal excretion [13]. As a consequence, 30–50% of patients on chronic glucocorticoid (GC) therapy may present fractures [13]. Moreover cumulative doses of GCs have been related to the risk of osteonecrosis in patients with ALL, particularly in older ones, showing a slower clearance of drugs and subsequent increased bone toxicity [14,15,16,17,18,19].

In adolescents suffering from ALL, important alterations of trabecular and cortical bone have been reported during the four-week induction treatment [20], while young adult survivors of pediatric ALL presented bone loss at the end of chemotherapy [21].

Moreover, osteoporosis and decreased bone mineral density (BMD) have been reported at the onset of childhood ALL, likely due to the action of autocrine factors linked to bone infiltration by malignant cells, indicating a direct effect of leukemic cells [22,23].

The possibility that leukemic B cells contribute directly to bone impairment has been explored in preclinical studies demonstrating a critical role of the receptor activator of the nuclear factor κB ligand (RANKL)-osteoprotegerin(OPG)/receptor activator of the nuclear factor κB (RANK) receptor axis in causing B-ALL cell-mediated bone damage [24,25].

All life long, the bone tissue has a continuous remodeling. The cells that promote bone remodeling are osteoclasts, the bone resorption cells, and osteoblasts, the bone-forming cells that lay down bone matrix proteins. The balance of bone remodeling is under control of the RANKL/OPG/RANK and Wnt/β-catenin pathways, which regulate osteoclastogenesis and osteoblastogenesis, respectively [26,27]. The Wnt/β-catenin pro-osteoblastogenic pathway can be inhibited by Dickkopf-1 (DKK-1) and sclerostin [28]. Pediatric bone diseases showed high serum levels of bone cytokines RANKL, sclerostin, and DKK-1. They also represent the target of emerging treatments for osteopenia and osteoporosis [29].

Markers of bone turnover reflect the activity of osteoblasts and osteoclasts [30]. Namely, bone alkaline phosphatase (bALP) and osteocalcin (OC), a non-collagenous marker, and procollagen I N-propeptide (PINP), a collagenous marker, reflect osteoblast activity. On the other side, the production of type I collagen degradation fragments amino terminal telopeptides of type I collagen (NTX) and carbossi terminal telopeptides of type I collagen (CTX) and the enzyme tartrate-resistant acid phosphatase type 5b (TRAcP5b) represent markers of bone resorption [31].

RANKL/RANK/OPG and Wnt/β-catenin pathways, which control osteoclastogenesis and osteoblastogenesis, regulate this equilibrium [26,27].

Scarce are data describing bone remodeling in children and adolescents with ALL. Gupta et al. evaluated, in children with ALL, a bone resorption marker (CTX) at diagnosis and at the 3-month time point during treatment, reporting its increase during treatment [32]. Boot et al. found low lumbar spine bone mineral density and low markers of bone turnover in children with ALL at diagnosis, hypothesizing a role of leukemia itself on bone [33].

The alterations of the bone marrow microenvironment during leukemogenesis seem to lead to perturbed hematopoiesis, impaired B-lymphopoiesis, reduced production of collagen, and reduced number of osteoblasts. Leukemia cells produced high levels of RANKL, sufficient to cause osteoclast-mediated bone resorption [24].

Previously, we observed a biomarker signature of bone resorption in children and adolescents under intensive chemotherapy treatment for ALL [34]. In the present study, we investigated the changes in bone remodeling and bone turnover markers at the diagnosis, before any treatment, and through intensive phases of therapy until stopping treatment, in a cohort of children and adolescent with ALL. This prospective longitudinal study, with 2 years of follow-up, aimed at clarifying the influence of leukemia itself and/or the disease treatment on bone impairment.

## 2. Results

### 2.1. Patient Population

Table 1 represents the clinical characteristics of the 22 ALL patients. According to the immunophenotype of blasts, B-lineage ALL was diagnosed in 19/22 subjects, T-lineage ALL in 2/22, and B-lineage ALL chromosome Philadelphia positive (ALL Ph+) in 1/22. Patients were treated according to AIEOP-BFM ALL 2017 and ESPhALL 2017 protocols [35]. A total of 8 patients were assigned to standard-risk protocols (8 patients with ALL B lineage), 13 to intermediate risk (11 ALL B lineage and 2 ALL T lineage), and 1 to high-risk ALL protocols. No relapse eventually occurred in the study population, nor did any patients receive bone marrow transplantation during the observation period. From the time point 1 onwards, all the patients showed complete remission of leukemia.

Bone pain at the onset of the disease was reported in 5/22 (22.7%) patients. In one patient out of 22 (4.5%), severe osteonecrosis was diagnosed during the maintenance phase of treatment.

### 2.2. Bone Turnover and Bone-Remodeling Markers

The variables 25-OH-vitamin D, CTX, OC, bALP, OPG, total RANKL, TRACP5b, P1NP, calcium, phosphorus, and PTH were collected and compared over time for all 22 patients. Comparisons between variables performed using an ANOVA on repeated measures were shown in Table 2. A comparison between controls and patients at diagnosis was performed using a parametric or non-parametric *t*-test. (Appendix A).

CTX, OC, P1NP, and bALP levels resulted significantly lower in the ALL population compared to controls (*p* = 0.009 for CTX and *p* < 0.001 for the others). The same trend was showed by DKK1 and sclerostin (*p* < 0.0001 and *p* = 0.023, respectively). On the contrary, RANKL ed OPG levels were higher in cases in respect to controls.

During different time points starting from diagnosis until stopping treatment, bone turnover and bone-remodeling markers showed increasing values; in particular, a great increase has been observed when comparing the T0 time point (diagnosis) to the T1 time point (end of induction chemotherapy). Less evident changes have been detected onwards (Table 2).

OH-vitamin D levels were below the referring range of normal values at diagnosis (T0) (insufficient levels) and dropped further to lower values during follow-up time points (T1–T4).

A multivariate approach by principal component analysis (PCA) has been used to discover changing in variables over the time and to identify patterns and correlations among them. Two preliminary tests have been performed to establish the feasibility of using a PCA. The first test was the Kaiser–Meyer–Olkin test (KMO test), which provided an overall acceptable value of 0.70; the second, the Bartlett test, provided a statistically significant result (*p* < 0.001). As presented in the scree plot, two principal components were fixed (Appendix A). The first principal component (PC1) and the second (PC2) explained 26.7 and 13.5% of overall variance, respectively, demonstrating a total variance equal to 40.2%. We investigated the importance of each variable building principal components. Notably, CTX, OC, and P1NP variables explained, each one, more than 60% of variance, followed by bALP, PTH, and 25-OH-vitamin D (about 50% of variance explained) and the others that did not explain much about variability over time (Appendix A). The observation of the loading plot confirmed such results. Variables with longer vectors had a strong weight in the principal components, explaining variability and therefore differences over time, while those with a shorter vector did not. Moreover, from the loading plot, correlations between variables were observed and, in particular, a strong correlation pattern among CTX, OC, and P1NP, and among TRACP5b and bALP, both in the negative direction of PC1, were found (Figure 1A, black and green circles).

Considering the score plot (Figure 1B), interestingly, a grouping between the time points (named from T0 to T4 and with a colors scale) along the negative direction of PC1 was found; instead, PC2 did not seem to affect separation between them. Two clusters were shown along the negative directions of PC1, moving from the onset point (blue color) to all other points. These results suggested that a marked increase in CTX, OC, and P1NP variables and a slight increase in the TRACP5b and bALP variables occurred during the follow-up, moving from the onset to the other time points.

Principal component analyses were also performed among all time points without the onset one (Appendix A). The results showed that two principal components explained 40.8% of total variance, but in this case, no separation or trend was found in the data set to underline that there are no differences in the bone-remodeling marker profile in a specific time point during the follow-up.

To strengthen the results shown by multivariate analyses, ANOVA on repeated measures provided strong statistically significant differences for the same variables. In particular, CTX values increased from 1.0 ± 0.5 ng/mL at the onset point to 2.5 ± 0.9 ng/mL at the third time point (*p* < 0.001); OC increased from 23.6 (17.1 ÷ 28.2) ng/mL at the onset point to 122.4 (97.4 ÷ 143.0) ng/mL at the third time point (*p* < 0.001); P1NP increased from 193.0 (117.7 ÷ 273.0) ng/mL at the onset point to 920.0 (812.2÷1050.2) ng/mL at the third time point (*p* < 0.001); TRACP5b and bALP showed an increase from the onset point to the third point of 10.0–18.2 U/L (*p* > 0.05) and 26.7–67.6 µg/L (*p* = 0.004), respectively. For CTX, OC, P1NP, and TRACP5b, a slight decrease, from time point 3 to the end of therapy, was observed (Figure 2 and Table 1).

The onset of disease proved to be a key point in the change of the bone remodeling and bone turnover markers. Indeed, the comparison of the CTX, OC, P1NP, and bALP values, between the controls and patients at the onset time point, revealed statistically significant lower levels of all of these variables (*p* < 0.01) and a trend towards lower levels (not significant) for TRACP5b with *p* = 0.308 (Appendix A).

## 3. Discussion

The skeleton is not an inert structure but continues to change throughout life. This process is known as bone remodeling, and it protects the structural integrity of the skeletal system and contributes metabolically to the balance of calcium and phosphorus. Remodeling involves the resorption of old or damaged bone, followed by the deposition of newly formed bone. Congenital or acquired diseases can alter bone remodeling, shifting it either towards a phase of excessive resorption or towards reduced bone formation [29,35]. In this study, we prospectively evaluated changes in bone remodeling and bone turnover markers in children with ALL, from the onset of the disease to during treatment.

Our findings demonstrated that at the diagnosis of leukemia, before any treatment, patients showed lower levels of bone turnover markers, both markers of bone formation, such as PINP and OC, and bone resorption, such as CTX, compared to healthy children.

This important result, not reported before, might represent both a direct and indirect effect of the leukemic disease itself. Leukemic blasts exert their effect on bone metabolism, likely due to the leukemia mass infiltrating the bone lacunae and to a crosstalk between lymphoblasts and bone, via the RANK/RANKL pathway, as hypothesized in preclinical studies [24,25]. A critical role of the RANK-RANKL axis causing B-precursor acute leukemia-mediated bone pathology has been postulated, since mouse and human B-lineage ALL cells have been demonstrated to produce RANKL and to cause bone destruction [24]. Moreover, in experimental models B-precursor ALL cells increase the activity of osteoclasts and produce high levels of RANKL while decreasing the number of osteoblastic cells in the bone marrow microenvironment [25]. Indeed, OC mRNA was significantly reduced in bone marrow stromal cells from leukemia mice [25].

In our study group of children with ALL, RANKL was significantly higher than in healthy children (Appendix A).

According to the prospective evaluation of bone turnover markers, an increase in CTX, OC, and P1NP and a slighter rise in TRACP5b and bALP has been documented during follow-up, moving from the onset of the disease to the following time points.

In particular, the change is more evident between the onset of the disease (T0) and the end of the induction phase (T1) (Table 2, Figure 2). At this time point, all the patients in our study had reached complete remission of the leukemia, which means that over 95% of bone marrow blasts have been destroyed. Starting from this point, coinciding with the clearance of malignant cells, an attempt of bone turnover recovery could be interpreted, driven by CTX, OC, and P1NP. However, observing different time points during follow-up, in particular looking at PCA, we detected a clusterization of samples taken at T1 to T4, apart from T0 (Figure 1).

This reflects the only slight differences in bone biomarkers during the follow-up time points: large changes occur from T0 to T1, while only minimal changes are evident from T1 to T4. An explanation for the observed data could be that leukemia itself acts as a driver of bone changes. Indeed, we found that there are no statistically significant differences in the bone-remodeling marker profile considering a specific time point during the later phases of treatment. Therefore, we might assume that great attention to bone health should be paid to the early phases of treatment, during the induction phase of chemotherapy when the clearance of the majority of blast cells eventually occurs.

In addition, chemotherapy and GCs did not seem to exacerbate the bone impairment differently from data reported by the current literature on bone health in children with leukemia [20,36]. However, the majority of studies have been conducted at the end of the induction phase of therapy, corresponding to the T1 time point in our study, which deserves the great changes in bone turnover. These changes persist throughout all treatment phases, until the last time point where an attempt to restore a balance of the bone remodeling phases is observed. To confirm this, CTX, OC, and P1NP, after the increase described at time point T1 and the slight increase up to T3, showed a decrease at T4, at the end of therapy, interpreted as a balance in active bone remodeling.

The poor impact of GCs on bone remodeling at different treatment phases in our study population could be explained by the fact that GC exposure is transient in children with ALL. According to the intermittent model of GC exposure that underlies current treatment protocols, bone remodeling can be recovered especially in growing children [37,38].

In our opinion, this recovery of bone remodeling balance restarts after the clearance of leukemic blasts from cancellous bone. Indeed, following the T0 starting time point, we did not document an imbalance of bone remodeling during treatment in our cohort of ALL children after the exposure to two phases of high-dose GC therapy, i.e., induction and reinduction or delayed intensification.

This observation is also confirmed by the results of our previous study where different doses of GC did not exert any significant, different effect on the bone marker profile of children after intensive phases of leukemia therapy [34]. In this context, we would like to underline the importance of endothelial and cardiovascular damage [6] reported early during leukemia treatment, which could influence the vascular damage to bone, which is responsible for osteonecrosis.

Another important player during the entire course of leukemia in children is the gut microbiome: its disruption influences immune responses and probably leukemia progression [39]. Through immune cross talk and inflammation, changes in gut microbiota composition might influence immune dysregulation, chronic inflammation, and bone health.

A further consideration deserves the 25-OH-vitamin D, which showed a statistically significant dropdown during treatment in our study population (Table 1). 25-OH-vitamin D at the insufficient or deficiency level is widely reported in the majority of children with leukemia, due to deficient oral intake and lifestyle. Therefore, a supplementation at higher levels in view of the bone health should be warranted [37].

## 4. Materials and Methods

### 4.1. Subjects

The study was designed as a prospective longitudinal study enrolling patients with diagnosis of ALL, followed through five different time points: at diagnosis, at the end of induction, at the start of reinduction, at the start of maintenance, and at the stop of therapy, which is scheduled two years later after the beginning of therapy.

In the study group, we included 22 children with a diagnosis of ALL (12 males), with a mean age at recruitment 5.5 ± 3.6 years. Patients received treatment according to the ongoing international AIEOP-BFM ALL protocols, which last 2 years. They were enrolled between June 2020 and September 2021, at the Pediatric Hematology and Oncology Clinic, University Hospital of Bari before starting any treatment. Blood samples were taken at diagnosis (time point 0) and at a further four time points, corresponding to different subsequent phases of chemotherapy, at the time of bone marrow regeneration, namely end of induction chemotherapy (time point 1), beginning of delayed intensification phase (time point 2), beginning of maintenance phase (time point 4) and stop therapy, and after stopping any antineoplastic treatment (time point 5). Inclusion criteria were as follows: (a) age of 1–18 years at the diagnosis of ALL and (b) first diagnosis of ALL (no relapsed disease).

Bone remodeling and bone turnover taken at diagnosis (time point 0) were compared to those of 49 healthy children (21 males, control groups) attending the University Pediatric Clinic of Bari for minor trauma (first aid) or allergology screening. Patients and controls were excluded from the study if they reported the use of vitamin and mineral supplements, the use of systemic GCs at any dosage before the onset of ALL, chronic diseases impacting bone metabolism (e.g., hypothyroidism or hyperthyroidism, Cushing’s syndrome, celiac disease, and anorexia nervosa), genetic syndromes, and fractures in the 6 months preceding the study.

The study protocol received approval by the Local Ethic Committee. Written informed consent was signed by parents or by patients > 18 years. All the procedures were in accordance with the guidelines of the Helsinki Declaration on Human Experimentation.

### 4.2. Risk Stratification and Treatment

Patients were stratified into three risk groups: standard, intermediate, and high risk, according to the immunophenotype, the molecular alterations of the leukemic blasts, and the response to treatment. Based on the intensity of the protocol, in the standard group we included patients with standard risk B-lineage ALL; in the intermediate group, we included medium risk B-lineage ALL and standard risk T-lineage ALL, and in the high-risk group we included high risk B- and T-lineage ALL and chromosoma Philadelphia positive ALL [40].

### 4.3. Clinical Data

Clinical characteristics of patients, such as age, sex, treatment, and leukemia classification were collected. Height, weight, and body mass index (BMI), as anthropometric parameters, were assessed. Anthropometric data were converted to age- and sex-matched standard deviation scores (SDS), using the national growth chart [41]. We recorded bone pain presented at the onset of the disease and the occurrence of osteonecrosis.

### 4.4. Bone Turnover Markers

A venous blood sample was taken from all participants at 08:00 a.m. after a 12 h night fast. Patients’ and controls’ samples were stored in aliquots at −20 °C for subsequent analysis, and measurements were made immediately after thawing. All measurement of serum bone markers were performed in the same assay run to avoid interassay variance.

Bone formation markers such as OC, P1NP, and bALP, as well as bone resorption markers such as CTX and TRACP5b, were assessed.

Serum OC, P1NP, bALP CTX, and TRAcP were measured by enzyme immunoassay, as previously descrbed [34]. The dosage of CTX, OC, TRACP5b, bALP, and P1NP was performed with a chemiluminescence assay using the TGSTA Technogenetics instrumentation (Techno-genetics, Milano, Italy).

Calcium and phosphorus concentrations were measured by the spectrophotometric method. Serum active intact parathyroid hormone (PTH) and 25(OH)-vitamin D were measured by immunological tests based on the principle of chemiluminescence as previously described [34].

All tests were performed in compliance with the manufacturer’s instructions and using suitable internal quality controls.

### 4.5. Bone Remodeling Cytokines Assessment

The analysis of RANKL, OPG, DKK-1, and Sclerostin were performed by enzyme immunoassays. The dosage was performed with an ELISA assay as previously described [34].

### 4.6. Statistical Analysis

Statistical analysis and data management were conducted using SigmaPlot 12.0 for Windows and the Chemometric Agile Tool (CAT). The normality of all numerical variables was assessed both graphically through box plots and statistically using the Shapiro–Wilk test. The homogeneity of variances was examined with Levene’s test. Variables following a normal distribution were reported as mean ± standard deviation, while those with non-normal distributions were summarized as median and interquartile ranges (IQRs).

Comparisons of variables were performed using either a parametric, one-way ANOVA for repeated measures or a non-parametric Friedmann test which compares ranked values across conditions over time. The one-way ANOVA and Friedmann test were followed by a Tukey and Holm–Sidak post hoc test for multiple comparisons, respectively. Moreover, a parametric t-test or non-parametric Mann–Whitney test to compare patients at onset with controls were conducted. A significance level of *p* < 0.05 was applied across all analyses.

To explore potential hidden patterns of association or correlation among variables and to evaluate their trends over time, a multivariate statistical approach was employed using principal component analysis (PCA). This technique serves as a powerful tool for identifying patterns of covariance among large sets of variables, which are often undetectable through traditional inferential statistics. The PCA method generates factors automatically through a standardized statistical process, with each factor explaining a specific portion of the overall variance in the data, represented by eigenvalues. Principal components are ranked in descending order based on the amount of variance they explain.

Interpreting PCA results requires two key visualizations: One of which is the loading plot, which reveals the significance of each original variable in forming the components and indicates whether the correlations between them are positive or negative. Variables located in the same region of the plot are positively correlated, while those in opposite quadrants show negative correlations. The analysis focuses on the components shown in the plot, with interpretations being more meaningful when the variance explained by the components is substantial. And the other is the score plot, which provides insights into the behavior of the data within the newly defined orthogonal space of the principal components, emphasizing similarities and differences between samples.

In this study, PCA was used to uncover trends over time, identify groupings, and detect specific patterns or distributions among variables. Before applying PCA, two preliminary tests were conducted to confirm its suitability. The first, the Kaiser–Meyer–Olkin (KMO) test, assessed sampling adequacy, with values above 0.70 considered appropriate for PCA. The second, the Bartlett test, evaluated the null hypothesis of no correlation among variables, yielding a statistically significant result of *p* < 0.001.

Additionally, data were standardized through column autoscaling (Z-score transformation) before running the PCA.

## 5. Conclusions

Our results suggest that the onset of leukemic disease has been revealed as a key point in determining a slowing of bone remodeling that represents in itself an index of well-being for the skeleton. Chemotherapy and GC therapy do not cause irreversible damage to skeletal health, as growing children have a considerable potential for recovery. It is therefore recommended to evaluate the skeletal health status in children with ALL at diagnosis, both through dosage of vitamin D and supplementation and through a detailed family history to detect factors that could aggravate the state of bone demineralization induced by the disease.

## Figures and Tables

**Figure 1 ijms-26-04307-f001:**
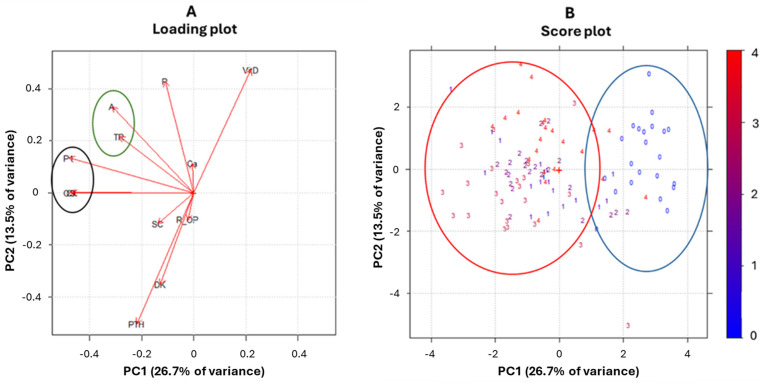
(**A**,**B**) Loading and score plots of PCA (principal components 1 and 2) of bone-remodeling markers for all patients. In the loading plot, black and green circles represent correlation patterns between variables. In the score plot, blue and red circles (and respective numbers) represent clusterization of the onset time point and the other time points during the follow-up, respectively. DK: dickkopf-related protein 1; R_OP: soluble receptor activator of nuclear factor-kappa B ligand-osteoprotegerin ratio; CX: C-terminal telopeptide cross-links of type I collagen; A: total alkaline phosphatase; TR: tartrate-resistant acid phosphatase; P1: procollagen type I N-terminal propeptide; OS: osteocalcin; Ca: calcium; P: phosphorus; PTH: parathormone; and SC: sclerostin.

**Figure 2 ijms-26-04307-f002:**
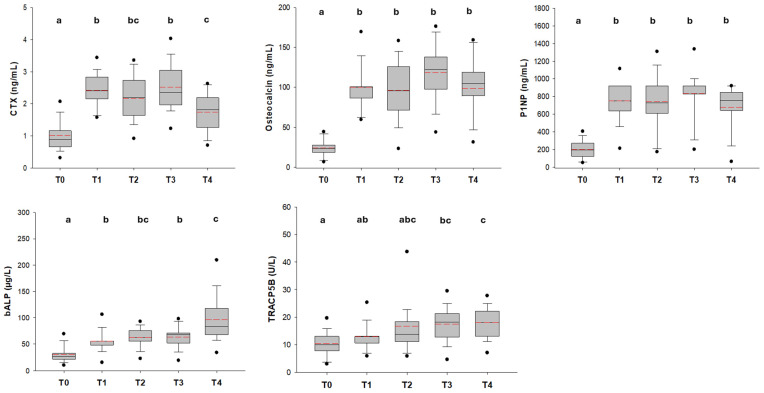
Distribution of bone-remodeling marker values, including of CTX, OC, P1NP, bALP, and TRACP5b, measured for all patients during the follow-up. The mean and median values are marked by a dotted red line and continuous black line, respectively. The different superscript letters in each box plot indicate a statistical difference between the time points (*p* < 0.05).

**Table 1 ijms-26-04307-t001:** Clinical characteristics of patients.

	Patients (n = 22)
Gender (male/female)	12/10
Age at diagnosis—years	5.5 ± 3.6
Height (SDS)	−0.75 ± 1.26
Weight (SDS)	−0.14 ± 1.18
BMI (SDS)	0.49 ± 0.91
Acute Lymphoblastic Leukemia phenotype:	
B-lineage	19
T-lineage	2
B-lineage t (9; 22)	1
Risk group:	
Standard risk	8
Intermediate risk	13
High risk	1
Bone pain yes/no	5/22

**Table 2 ijms-26-04307-t002:** Bone turnover and bone-remodeling markers at different longitudinal time points.

Marker	T0	T1	T2	T3	T4	*p*-Value
CTX-I (ng/mL)	1.0 ± 0.5 ^a^	2.4 ± 0.6 ^b^	2.2 ± 0.7 ^bc^	2.5 ± 0.9 ^b^	1.7 ± 0.6 ^c^	**^1^ <0.001**
OC (ng/mL)	23.6 (17.1 ÷ 28.2) ^a^	100.0 (84.6 ÷ 103.0) ^b^	95.9 (71.4 ÷ 127.9) ^b^	122.5 (97.4 ÷ 143.0) ^b^	104.7 (79.8 ÷ 121.4) ^b^	**^2^ <0.001**
P1NP (ng/mL)	193.0 (117.7 ÷ 273.0) ^a^	749.0 (634.0 ÷ 920.0) ^b^	728.4 (606.3 ÷ 920.0) ^b^	920.0 (812.2 ÷ 1050.2) ^b^	753.9 (640.0 ÷ 860.7) ^b^	**^2^ <0.001**
bALP (µg/L)	26.7 (21.4 ÷ 32.3) ^a^	55.7 (48.0 ÷ 56.0) ^b^	62.2 (55.2 ÷ 76.4) ^bc^	67.6 (51.9 ÷ 71.3) ^b^	83.5 (67.8 ÷ 119.7) ^c^	**^2^ <0.001**
TRAcP5b (U/L)	10.0 (7.6 ÷ 13.0) ^a^	12.9 (10.5 ÷ 13.0) ^ab^	13.9 (11.0 ÷ 18.7) ^abc^	18.2 (12.6 ÷ 21.5) ^bc^	18.0 (13.0 ÷ 22.2) ^c^	**^2^ <0.001**
25-OH-Vit D	25.5 (19.0 ÷ 35.3) ^a^	15.0 (11.5 ÷ 19.3) ^b^	15.0 (10.0 ÷ 21.0) ^b^	12.0 (7.0 ÷ 16.3) ^b^	27.0 (20.8 ÷ 32.3) ^a^	**^2^ <0.001**
RANKL(pmol/L)	4302 (1312 ÷ 20,336) ^a^	5859 (1315 ÷ 8383) ^a^	3264 (562 ÷ 13,859) ^a^	1601 (843 ÷ 13,014) ^a^	2010 (86 ÷ 24600) ^a^	**^2^** 0.220
OPG (pmol/L)	4.4 (2.9 ÷ 7.5) ^ab^	3.8 (3.5 ÷ 4.1) ^b^	3.1 (2.7 ÷ 3.9) ^c^	3.4 (2.8 ÷ 4.2) ^bc^	5.1 (4.0 ÷ 6.3) ^a^	**^2^ <0.001**
RANKL/OPG ratio	920 (311 ÷ 2839) ^a^	1469 (263 ÷ 2212) ^a^	882 (161 ÷ 3873) ^a^	620 (178 ÷ 4089) ^a^	365 (15 ÷ 3756) ^a^	**^2^** 0.567
PTH (pg/mL)	18.0 (8.0 ÷ 29.3) ^a^	40.0 (32.8 ÷ 42.5) ^ab^	46.0 (36.5 ÷ 52.5) ^b^	55.0 (39.0 ÷ 70.5) ^b^	32.5 (21.0 ÷ 35.5) ^a^	**^2^ <0.001**
DKK-1 (pg/mL)	2266 (1362 ÷ 4902) ^a^	4724 (3416 ÷ 5752) ^ab^	4016 (2752 ÷ 4814) ^a^	6180 (4564 ÷ 8405) ^b^	3346 (2765 ÷ 4191) ^a^	**^2^ <0.001**
Sclerostin (pmol/L)	14.9 (13.9 ÷ 18.3) ^a^	17.6 (15.7 ÷ 19.9) ^a^	16.7 (15.3 ÷ 22.3) ^a^	19.0 (13.9 ÷ 23.2) ^a^	18.5 (16.6 ÷ 20.6) ^a^	**^2^** 0.162
Calcium (mg/dL)	9.1 (8.6 ÷ 9.4) ^a^	9.4 (8.8 ÷ 9.5) ^ab^	9.1 (8.8 ÷ 9.7) ^ab^	9.3 (8.8 ÷ 9.4) ^a^	9.8 (9.4 ÷ 10.1)^b^	**^2^ 0.021**
Phosphorus (mg/dL)	5.2 (4.4 ÷ 5.6) ^a^	5.0 (4.7 ÷ 5.1) ^a^	5.0 (4.8 ÷ 5.2) ^a^	5.0 (4.5 ÷ 5.3) ^a^	4.9 (4.1 ÷ 5.1) ^a^	**^2^** 0.517

Data are presented as mean ± SD for normally distributed variables and median with interquartile range for non-normally distributed variables. CTX: C-terminal telopeptide cross-links of type I collagen; OC: osteocalcin; P1NP: procollagen type I N-terminal propeptide; bALP: bone alkaline phosphatase; TRAcP5b: tartrate-resistant acid phosphatase; 25-OH-Vit D: 25-hydroxy-vitamin D; RANKL: soluble receptor activator of nuclear factor kappa-B ligand; OPG: osteoprotegerin; PTH: parathormone; DKK-1: dickkopf-related protein 1. Different superscript letters above each value, in each line, indicate statistically significant differences between the time points. The significance level for ANOVA was set at 0.05 in all cases. ^1^ Parametric One-way ANOVA on repeated measures. ^2^ Non-parametric One-way ANOVA on repeated measures.

## Data Availability

The data presented in this study are available on request from the corresponding author.

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
