# Peer review of "Bone Remodeling in Children with Acute Lymphoblastic Leukemia: A Two-Year Prospective Longitudinal Study"

_ijms, 2025, doi:10.3390/ijms26094307_

Round 1

Reviewer 1 Report

Comments and Suggestions for Authors

This 2-year prospective longitudinal study investigated bone remodeling and bone turnover markers at diagnosis, during treatment until stopping treatment, in ALL patients < 18 years, to clarify the influence of leukemia itself and/or chemotherapy on bone. The work is OK with me.

The topic is relevant,  however it is not the first publication on this topic. Deserves to be published.

Compared with other published material, this paper gives more evidence concerning bone turnover in paediatric oncology.

Regarding the methodology, a larger patient population in the future, however concerning this  study, the methodology is OK.

Conclusions are consistent with the evidence and arguments presented.

References are appropriate. Historically, there have been several relevant publication concerning  bone metabolism (especially bone turnover markers) in children with ALL.

For example:

Hoorweg-Nijman JJ et al.  .Bone mineral density and markers of bone turnover in young adult survivors of childhood lymphoblastic leukaemia. Clin Endocrinol (Oxf). 1999 Feb;50(2):237-44.

Boot AM et al. Bone mineral density in children with acute lymphoblastic leukaemia. Eur J Cancer. 1999 Nov;35(12):1693-7.

Gupta V et al. Alterations in Bone Turnover during Chemotherapy in Children with Acute Lymphoblastic Leukemia. South Asian J Cancer. 2021 Dec 20;10(3):183-186.

The list of references  is  really up-to-date, but maybe adding some of the above mentioned citations  could  be beneficial.

Furthermore, page 10-11, lines 400 – 414 :  is this text necessary?  Seems to me as  “copy and paste“ from the Instructions to Authors.

Comments on the Quality of English Language

Page 2, line 74: I´d prefer “bone resorption“  to   “bone reabsorption“

Page 2, line 82:  replace   “work“  with  “activity“

Page 2, line 92: replace “led“  with “lead“

Page 3, line 125: better replace “showed“ with “shown“

Author Response

Reviewer 1

This 2-year prospective longitudinal study investigated bone remodeling and bone turnover markers at diagnosis, during treatment until stopping treatment, in ALL patients < 18 years, to clarify the influence of leukemia itself and/or chemotherapy on bone. The work is OK with me.

The topic is relevant,  however it is not the first publication on this topic. Deserves to be published.

Compared with other published material, this paper gives more evidence concerning bone turnover in paediatric oncology.

Regarding the methodology, a larger patient population in the future, however concerning this  study, the methodology is OK.

Conclusions are consistent with the evidence and arguments presented.

References are appropriate. Historically, there have been several relevant publication concerning  bone metabolism (especially bone turnover markers) in children with ALL.

For example:

Hoorweg-Nijman JJ et al.  .Bone mineral density and markers of bone turnover in young adult survivors of childhood lymphoblastic leukaemia. Clin Endocrinol (Oxf). 1999 Feb;50(2):237-44.

Boot AM et al. Bone mineral density in children with acute lymphoblastic leukaemia. Eur J Cancer. 1999 Nov;35(12):1693-7.

Gupta V et al. Alterations in Bone Turnover during Chemotherapy in Children with Acute Lymphoblastic Leukemia. South Asian J Cancer. 2021 Dec 20;10(3):183-186.

The list of references  is  really up-to-date, but may be adding some of the above mentioned citations  could  be beneficial.

- we thank the reviewer for this comment. Two papers from Gupta et al and Boot et al have been added to the reference list and cited in the introduction.

Furthermore, page 10-11, lines 400 – 414 :  is this text necessary?  Seems to me as  “copy and paste“ from the Instructions to Authors.

- thank you to the reviewer: this part has been removed, it was a mistake leaving it in the manuscript

Comments on the Quality of English Language

Page 2, line 74: I´d prefer “bone resorption“  to   “bone reabsorption“

Page 2, line 82:  replace   “work“  with  “activity“

Page 2, line 92: replace “led“  with “lead“

Page 3, line 125: better replace “showed“ with “shown

Thank you for the comment. Changes have been done.

Reviewer 2 Report

Comments and Suggestions for Authors

Recommendations:

  1. The study has very high percent match needs radical reduction.
  2. what ANOVA test was used? the methods section needs to reveal.
  3. I did not see Kendall's Tau test, which is most accurate to assess trend of paired samples over time. So I cannot consider the statistics really reliable.
  4. Discuss novelty factors that can influence the complications of leukemia: https://doi.org/10.3390/children12020166
  5. I do not see a pattern rather than the bone resorption increases comparative to baseline, you mentioned here: To confirm this, CTX, OC and P1NP, after the increase described 256
    at time point T1 and the slight increase up to T3, showed a decrease at T4, at the end of 257 therapy, interpreted as a balance in active bone remodeling.-are this changes statistically significant? I did not see any post-hoc test for this non-parametric ANOVA (Wilcoxon or Conover's test) to prove that this changes are statistically significant! 
Comments on the Quality of English Language

English fine!

Author Response

Reviewer 2

  1. The study has very high percent match needs radical reduction. &nbsp&nbsp&nbsp&nbsp&nbsp- thank you very much for this comment. The percentage of plagiarism has been greatly reduced. Changes have been made, and highlighted in green in the manuscript.
  2. what ANOVA test was used? the methods section needs to reveal. Dear reviewer, thank you for your question. For variables that followed a normal distribution of the data, we applied an ANOVA on repeated measures followed by Tukey post hoc test for multiple comparisons. In cases where the normality assumption was not met, we used the non-parametric Friedman test, which compares ranked values across conditions followed by the Holm-Sidak post hoc test, a good method for conducting post-hoc comparisons between groups, maintaining control of the Type I error more efficiently than Bonferroni.
  1. I did not see Kendall's Tau test, which is most accurate to assess trend of paired samples over So I cannot consider the statistics really reliable.     Dear reviewer, thank you for your valuable suggestion. However, I respectfully disagree with this observation, and I will explain my reasons. The Kendall’s Tau test is better suited for analyzing the correlation between two specific variables over time and is classified as a univariate test. On the other hand, when dealing with multiple temporal variables, PCA becomes more powerful and is the preferred method. PCA is highly effective in time-dependent datasets because it helps identify dominant trends and reduces dimensionality, all while retaining the essential patterns over time. Rather than directly analyzing raw time series data, PCA extracts principal components that capture the most significant variations in the dataset. It is particularly advantageous for multivariate time data, allowing for the efficient handling of several time-dependent variables. In medical research, for example, PCA can be extremely useful for studying patient health trends over time. Therefore, the choice between methods should be based on what is statistically most effective, rather than just perceived reliability.
  2. Discuss novelty factors that can influence the complications of leukemia: https://doi.org/10.3390/children12020166 Thank you for this interesting comment. A sentence has been added at the end of the discussion to comment the emerging influence of gut microbiome and dysbiosis on childhodd leukemia.
  3. I do not see a pattern rather than the bone resorption increases comparative to baseline, you mentioned here: To confirm this, CTX, OC and P1NP, after the increase described 256
    at time point T1 and the slight increase up to T3, showed a decrease at T4, at the end of 257 therapy, interpreted as a balance in active bone remodeling.-are this changes statistically significant? I did not see any post-hoc test for this non-parametric ANOVA (Wilcoxon or Conover's test) to prove that this changes are statistically significant!  Dear reviewer, thank you for your comment. The statistical differences (significant or not) between times T3 and T4, as well as between all other time points, were assessed through post hoc tests that we have added in the Materials and Methods section, as requested by the reviewer. The evidence of the comparisons was already present through the superscript letters in the box plots in Figure 2.